# Light affects behavioral despair involving the clock gene *Period 1*

Iwona Olejniczak[1⍟], Jürgen A. Ripperger[1⍟], Federica Sandrelli[2], Anna Schnell[1], Laureen Mansencal-Strittmatter[1], Katrin Wendrich[1], Ka Yi Hui[1], Andrea Brenna[1], Naila Ben Fredj[1], Urs Albrecht[1]*

**1** Department of Biology, University of Fribourg, Fribourg, Switzerland, **2** Department of Biology, University of Padova, Padova, Italy

⍟ These authors contributed equally to this work.
\* urs.albrecht@unifr.ch

**Data Availability Statement:** All RNA sequencing files are available from the SRA database (accession number PRJNA628975).

## Abstract

Light at night has strong effects on physiology and behavior of mammals. It affects mood in humans, which is exploited as light therapy, and has been shown to reset the circadian clock in the suprachiasmatic nuclei (SCN). This resetting is paramount to align physiological and biochemical timing to the environmental light-dark cycle. Here we provide evidence that light at zeitgeber time (ZT) 22 affects mood-related behaviors also in mice by activating the clock gene *Period1 (Per1)* in the lateral habenula (LHb), a brain region known to modulate mood-related behaviors. We show that complete deletion of *Per1* in mice led to depressive-like behavior and loss of the beneficial effects of light on this behavior. In contrast, specific deletion of *Per1* in the region of the LHb did not affect mood-related behavior, but suppressed the beneficial effects of light. RNA sequence analysis in the mesolimbic dopaminergic system revealed profound changes of gene expression after a light pulse at ZT22. In the nucleus accumbens (NAc), sensory perception of smell and G-protein coupled receptor signaling were affected the most. Interestingly, most of these genes were not affected in *Per1* knock-out animals, indicating that induction of *Per1* by light serves as a filter for light-mediated gene expression in the brain. Taken together we show that light affects mood-related behavior in mice at least in part via induction of *Per1* in the LHb with consequences on mood-related behavior and signaling mechanisms in the mesolimbic dopaminergic system.

## Author summary

Day-length has a profound effect on mood status in humans. Short winter days are often associated with seasonal affective disorder, which is a form of depression. Light therapy can alleviate the symptoms of this disorder, but the mechanisms how light can do this are unknown. Using mice as a model system mimicking the effects of light on depressive behavior in humans, we found that the clock gene *Period 1* is an important component necessary to mediate beneficial light effects on depressive behavior.

**Funding:** This work was supported by the Velux Foundation (https://veluxstiftung.ch) Projects 995 and 772 to U.A. and the Swiss National Science Foundation (http://www.snf.ch) project number 310030_184667/1. The funders had no role in study design, data collection and analysis, decision to publish, or preparation of the manuscript.

**Competing interests:** The authors have declared that no competing interests exist.

## Introduction

The circadian clock has evolved from cyanobacteria to humans in response to the daily light-dark cycle [1]. The internalization of the regularly recurring alternation of light and darkness allowed organisms to anticipate this change. This enabled them to prepare biochemical and physiological processes in order to optimally respond to the upcoming daily challenges and increase survival in a competitive environment. Malfunctioning or disruption of the circadian clock system results in mammals in various pathologies including obesity, cancer, and neurological dysfunctions [2]. For the maintenance of synchronicity within the mammalian body and with the environmental light-dark cycle, the suprachiasmatic nuclei (SCN) receive light information directly from intrinsically photosensitive retinal ganglion cells (ipRGCs) [3–5]. This information is converted by the SCN into humoral and neuronal signals to set the phase of circadian oscillators and drive circadian rhythmic coherence in the body [6].

At the cellular level, the clock mechanism is relying on feedback loops involving a set of clock genes. In mammals the main feedback loop consists of two *period* (*Per1* and *Per2*) and two *cryptochrome* genes (*Cry1* and *Cry2*), whose transcription is controlled by the transcriptional activators BMAL1 and CLOCK (or NPAS2). PER and CRY form complexes with additional proteins, enter the nucleus and inhibit the activity of BMAL1/CLOCK complexes stopping their own activation. A second feedback loop involving the nuclear orphan receptors REV-ERB (α, β) and ROR (α, β, γ) regulates the expression of the *Bmal1* and *Clock* genes, whose proteins in turn regulate the *Rev-erb* and *Ror* genes [7]. Posttranslational modifications modulate the activity and stability of clock proteins and thereby contribute to and fine-tune circadian rhythm generation [8].

In order to adjust the clock to environmental signals such as light, at least one of the clock components needs to be responsive to the stimulus. This leads to a shift of clock phase in order to synchronize the organism to environmental time. Interestingly, expression of the *Per1* gene is inducible in the SCN by a nocturnal light pulse at zeitgeber time (ZT) 22 [9,10]. At this time point, light not only causes eventual adaptation of the mammalian circadian clock to a new time zone but is also most effective in the treatment of some forms of depression, such as seasonal affective disorder [11]. That light exerts powerful effects on mood and cognition has been documented not only in humans [12–14], but also in laboratory animals [15–17]. The neural basis for the influence of light on mood and learning appear to be distinct retina-brain pathways involving ipRGCs that either project to the SCN to influence learning or to the perihabenular nucleus (PHb) in the thalamus to modulate mood [18]. Furthermore, anti-depressive effects of light therapy require activation of a pathway leading from the retina via the ventral lateral geniculate nucleus (vLGN)/intergeniculate leaflet (IGL) to the lateral habenula (LHb) [19].

Here we find that light induces *Per1* gene expression in the LHb and suppression of it increases immobility in the FST. Furthermore, lack of *Per1* in mice increases the number of light-induced genes 10 hours later in the SCN and LHb, while decreasing this number in the NAc. The profound changes in gene expression in the mesolimbic dopaminergic system are linked to sensory perception of smell and G-protein coupled receptor signaling. Our observations suggest an involvement of *Per1* in the light-mediated pathway that regulates mood-related behaviors.

## Results

### Light at ZT22 affects despair-based behavior

In order to test the effect of light on mood-related behavior in mice, we applied a 30-min polychromatic light pulse (white light) to animals that were kept in a 12-hour light/12-hour dark

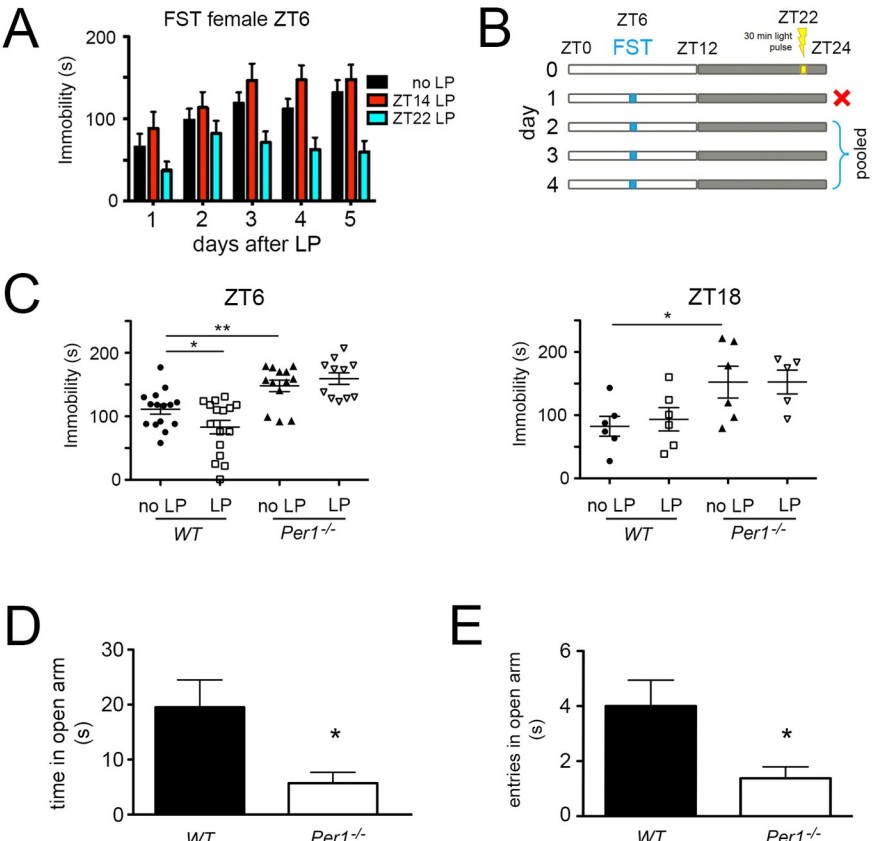

**Fig 1. Light at ZT22 affects despair-based behavior.** (A) Immobility time in the forced swim test (FST) of wild type female mice assessed over several days at ZT6 after no light pulse (LP) (black bars), after a LP at ZT14 (red bars), and after a LP at ZT22 (blue bars). Two-way repeated measures ANOVA (n = 13–15), ZT14 LP p = 0.255, ZT22 LP p = 0.014, values are means ± SEM (B) Light pulse treatment and assessment protocol using the FST. The first day after the light pulse the FST was performed but only the data from days 2–4 were pooled and used for further analysis. (C) Immobility time of wild type and *Per1*[-/-] female mice with and without LP at ZT22 assessed at ZT6 (left panel, n = 11–16) or ZT18 (right panel, n = 5–6). Unpaired t-test, *p<0.05, **p<0.01, values are means ± SEM. (D) Time spent in the open parts of an O-maze. Unpaired t-test, n = 8 for each genotype, *p<0.05, values are means ± SEM. (E) Entries into the open parts of an O-maze. Unpaired t-test, n = 8 for each genotype, *p<0.05, values are means ± SEM.

(12:12 LD) cycle. The light pulse was applied in the dark phase at zeitgeber time (ZT) 14 or 22, with ZT0 being lights on and ZT12 lights off. The mice were then subjected to a forced swim test (FST) at ZT6 for the next five days where their immobility time was assessed (Figs 1A and S1A). Animals that received no light pulse (black bars) or a light pulse at ZT14 (red bars) showed comparable immobility times (Figs 1A and S1A). In contrast, mice that received a light pulse at ZT22 (blue bars) displayed lower immobility times compared to control animals. This difference was statistically significant in females (Fig 1A), but only showed a similar trend in males (S1A Fig), which is consistent with previous findings in rats [17].

Based on these data we performed the subsequent experiments with female mice and applied a light pulse only at ZT22. The FST was performed at ZT6 and immobility time was assessed for the four following days. The data of days 2–4 were pooled to compare immobility times between animals (Fig 1B). Consistent with the results in Fig 1A wild type mice showed reduced immobility time at ZT6 when they received a light pulse at ZT22 (Fig 1C, left panel), but not when they were assessed at ZT18 (Fig 1C, right panel). Since *Per1* is a light-inducible gene in the SCN [9,10], we tested mice lacking this gene in the same paradigm. At ZT6, as well

as at ZT18, *Per1* knock-out (*Per1*$^{-/-}$) mice showed increased immobility time compared to wild type animals, and a light pulse did not affect their immobility time in the FST (Fig 1C). Hence, *Per1*$^{-/-}$ animals showed an increase in behavioral despair, which was not decreased by light in contrast to wild type animals. Next, we used the sucrose preference test, which provides a measure for anhedonia, another characteristic of depression (decreased ability to experience pleasure; [20]). We observed no difference between wild type and *Per1*$^{-/-}$ mice in this experiment (S1B Fig). To test anxiety-related behavior in the two genotypes we used the elevated O-maze test [20]. The *Per1*$^{-/-}$ mice spent less time in the open area and did enter it significantly less compared to wild type control animals (Fig 1D and 1E). These results indicated that *Per1*$^{-/-}$ animals were more anxious than controls.

## Light at ZT22 induces *Per1* in the lateral habenula

Because the LHb is involved in the regulation of the behavioral response of mice in the FST [19], we performed in situ hybridization experiments on mouse brain sections in order to evaluate, whether the *Per1* and *Per2* genes were expressed in the LHb and the medial habenula (MHb). We observed, that *Per1* was expressed in both the LHb and MHb although with a much lower amplitude than in the SCN (Fig 2A). *Per2* mRNA was expressed in both the LHb and the MHb but to a much lower extent (Fig 2B). Since light at ZT22 affects immobility in the FST (Fig 1C), and induces *Per1* but not *Per2* expression in the SCN [9,10] we tested the effect of a light pulse at ZT22 on expression of the *Per1* gene. We detected strong induction of *Per1* in the SCN one hour after the light pulse, which was weaker after two hours (Fig 2C, left panel) consistent with previous findings [9,10]. A similar pattern of *Per1* gene induction was observed in the LHb although this was less pronounced when compared to the SCN (Fig 2C, right panel). Interestingly, a light pulse at ZT14 did not induce *Per1* in the LHb contrary to the SCN (S2A Fig) paralleling the absence of a light effect in the FST (Fig 1A).

To corroborate our observations, we performed quantitative real-time PCR analysis. In order to demonstrate the accuracy of the isolation of SCN, LHb, VTA, and NAc tissue from wild type mouse brains, we used markers specific to the corresponding brain region (S2B Fig). Mice that received a light pulse at ZT22 were compared to controls not receiving any light. We detected induction of *Per1* but not *Per2* in the SCN, the LHb and the VTA one hour after the light pulse (Fig 2D), which was consistent with the data we obtained by using in situ hybridization. Taken together our data suggest a role of *Per1* in the light-mediated effects on despair related behavior and that this may involve the LHb.

## Deletion of neuronal *Per1* in the area of the lateral habenula affects light mediated effects on despair

In order to challenge the hypothesis that expression of *Per1* in the LHb plays an important role in the regulation of despair based and anxiety-related behavior, we deleted *Per1* in neurons of the LHb. To this end we generated mice with a floxed *Per1* allele (Fig 3A). Deletion of exons 4–16 resulted in a reading frame that could only be resumed in exon 22, thereby deleting almost the entire gene. The various alleles could be analyzed by PCR resulting in a 280 bp amplicon for the wild-type allele (wt), a 468 bp amplicon for the floxed allele (fl), and a 585 bp amplicon for the *Per1* deleted allele (*Per1*$^{-}$, Dfl) (Fig 3A and 3B). To ensure efficient delivery of the Cre recombinase into the LHb we tested the transduction efficiency and tissue penetration of various serotypes of adeno-associated virus (AAV) expressing EGFP (S3A Fig). We injected AAV6 containing expression vectors for either a control construct that expresses EGFP under the synapsin promoter or the same construct additionally expressing iCre into the LHb (Fig 3C). We observed that presence of CRE (red color) led to strong reduction of PER1

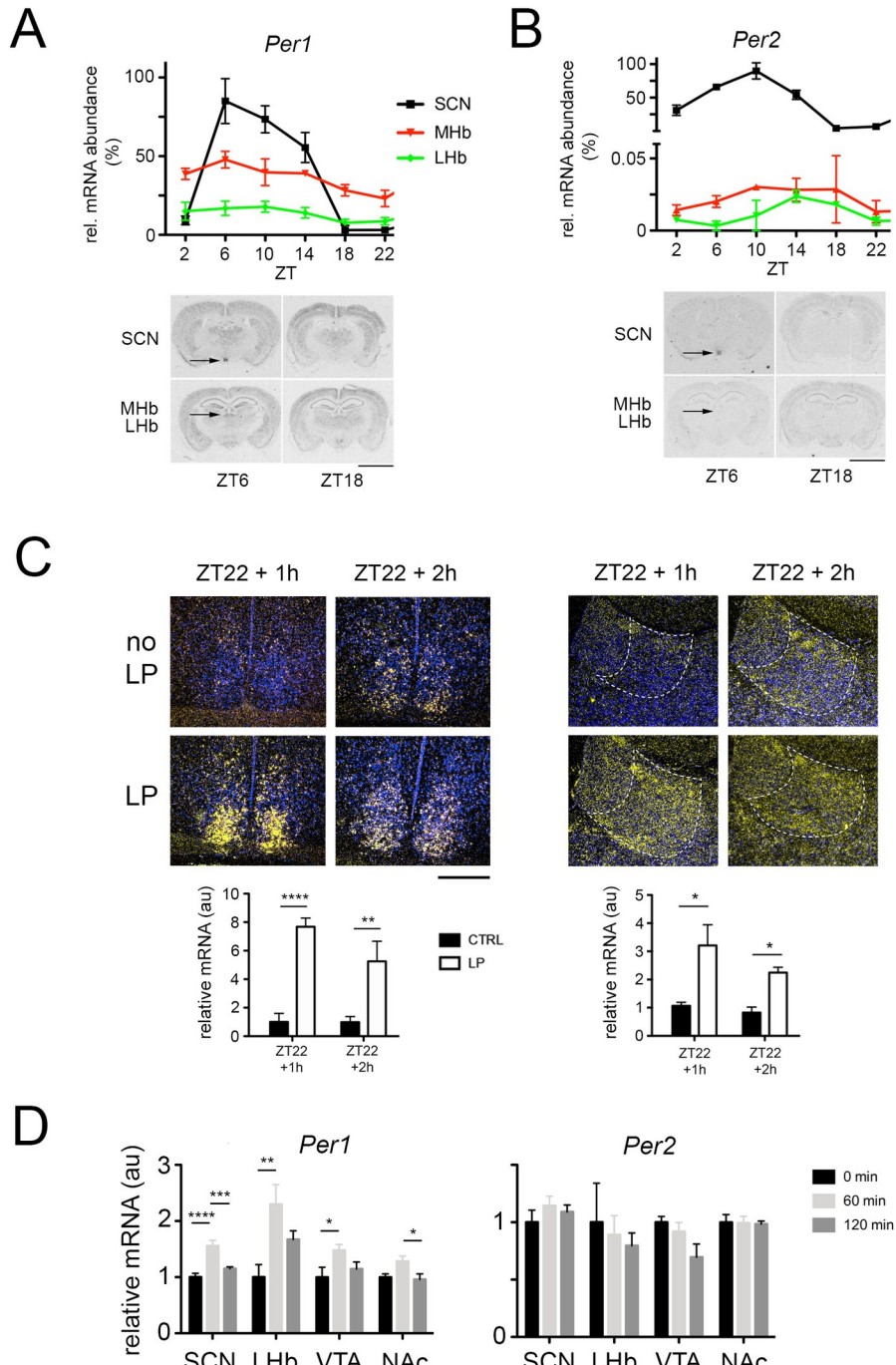

**Fig 2. Light induces the clock gene *Per1* in the lateral habenula of mice.** In situ hybridization revealing expression of *Per1* (A) or *Per2* (B) in the SCN, the medial habenula (MHb) and the lateral habenula (LHb). The top panel shows the quantification of *Per1* expression in the SCN, the MHb and the LHb, values are means ± SEM. CircWave analysis revealed circadian expression of both genes in the tissues shown (n = 3, p<0.05). The bottom panel shows representative images of brain sections at ZT6 and ZT18 with expression signal (black) of *Per1* or *Per2* in the respective brain regions. Scale bar: 5 mm. (C) Induction of *Per1* mRNA expression after a LP at ZT22 in the SCN (left panels) and in the lateral habenula (right panels). The blue color depicts cell nuclei (Hoechst staining) and the yellow color shows radioactively labeled antisense *Per1* riboprobe hybridized to *Per1* mRNA. Below the photo-micrographs quantification is shown. Two-way ANOVA with Bonferroni multiple comparisons test, n = 3, *p<0.05, **p<0.01, ****p<0.0001, values are means ± SEM. Scale bar = 200 μm. (D) Quantitative PCR revealed mRNA expression levels of *Per1* (left panel) and *Per2* (right panel) in the suprachiasmatic nuclei (SCN), lateral habenula (LHb), ventral tegmental area (VTA) and nucleus accumbens (NAc) after 60 min. (light grey bars) and 120 min. (dark grey bars) of a light pulse at

ZT22 (0 min., black bars). *Per1* mRNA was induced in the SCN, LHb, and VTA, whereas *Per2* was not induced in any of the tissues investigated. One-way ANOVA, Tukey's multiple comparisons test, n = 12–16, *p<0.05, **p<0.01, ***p<0.001, ****p<0.0001, values are means ± SEM.

immunohistochemistry signal (pink color), indicating successful *Per1* gene deletion. After recovery from the procedure the animals were tested for efficiency of *Per1* deletion. A significant decrease of *Per1* expression was observed in the LHb but not in the SCN (Fig 3D, tissue control S3B Fig). The remaining *Per1* mRNA detected in the LHb probably stemmed from *Per1* expression in other cells than neurons, such as glial cells. In line with these results was the significant reduction of PER1 protein in the LHb after deletion of *Per1* (Fig 3E). Animals injected with the virus expressing iCre, termed vS*Per1*, or with the control virus, were subjected to the FST at ZT6. Mice that received the control virus still responded to a light pulse at ZT22 with a reduction of immobility, whereas the vS*Per1* animals did not respond to the light pulse (Fig 3F). Interestingly, deletion of *Per1* in the LHb was not sufficient to increase immobility time in the FST (Fig 3F), which is in contrast to the total body *Per1* knock-out (Fig 1C). We also evaluated the vS*Per1* animals in the elevated O-maze test. Compared to the control mice these animals spent a similar amount of time in the open arm (Fig 3G). Overall these results indicate that *Per1* expression in neurons of the LHb is highly important for mediating the light effects observed in the FST. However, deletion of *Per1* in the LHb is not sufficient to affect immobility time in the FST without a light pulse, nor the time spent in the open arm of the O-maze. This suggests that *Per1* expression in other brain regions is involved in the phenotypes observed in Fig 1.

## Light induction of PER1 protein shapes the transcriptome in the brain in a regional manner

Presence or absence of the *Per1* gene changes behavior of mice in the FST. In order to relate this behavioral effect to alterations in the transcriptome, we performed RNA sequencing experiments. Because we were interested in the downstream impact of *Per1*, we first studied the amount of PER1 protein after the light pulse at ZT22 in the SCN and the LHb. Previous studies showed that after a light pulse at ZT22 murine PER1 protein in the SCN was increased 10 hours later (ZT8) when compared to mice that received no light [21]. We confirmed this observation in the SCN by Western blot and found a similar increase of PER1 protein in the LHb (Fig 4A). This suggested that *Per1* is light-inducible in the LHb.

Because PER1 protein presence and hence its biological function is increased at ZT8 after a light pulse at ZT22 compared to controls, we collected brain tissues at ZT8. We chose the SCN as reference tissue and the LHb, VTA and NAc as tissues known to be involved in the regulation of behaviors related to mood and despair as part of the mesolimbic dopaminergic system [19,22]. The tissues from wild type and *Per1*$^{-/-}$ mice receiving no light or light at ZT22 were isolated at ZT8. We validated the specificity of the tissues isolated using marker genes such as lrs4 (SCN), Gpr151 (LHb), Tacr3 (VTA) and Tac1 (NAc) (S4 Fig) that were determined from the Allan brain atlas and normalized according to [23]. RNA sequencing was performed and the volcano plots comparing gene expression under no light versus light at ZT22 for the various tissues and genotypes are shown in Fig 4B. In wild type mice the SCN and LHb showed several genes are up- or down-regulated 10 hours after the light pulse (black dots) with the VTA displaying a much lower number and the NAc showing a massive number of affected genes. In contrast, the change in gene expression is opposite in the *Per1*$^{-/-}$ mice. A large number of genes affected by light were observed in the SCN and LHb with low numbers in the VTA and the NAc. These results are consistent with the role of PER1 as a regulator of

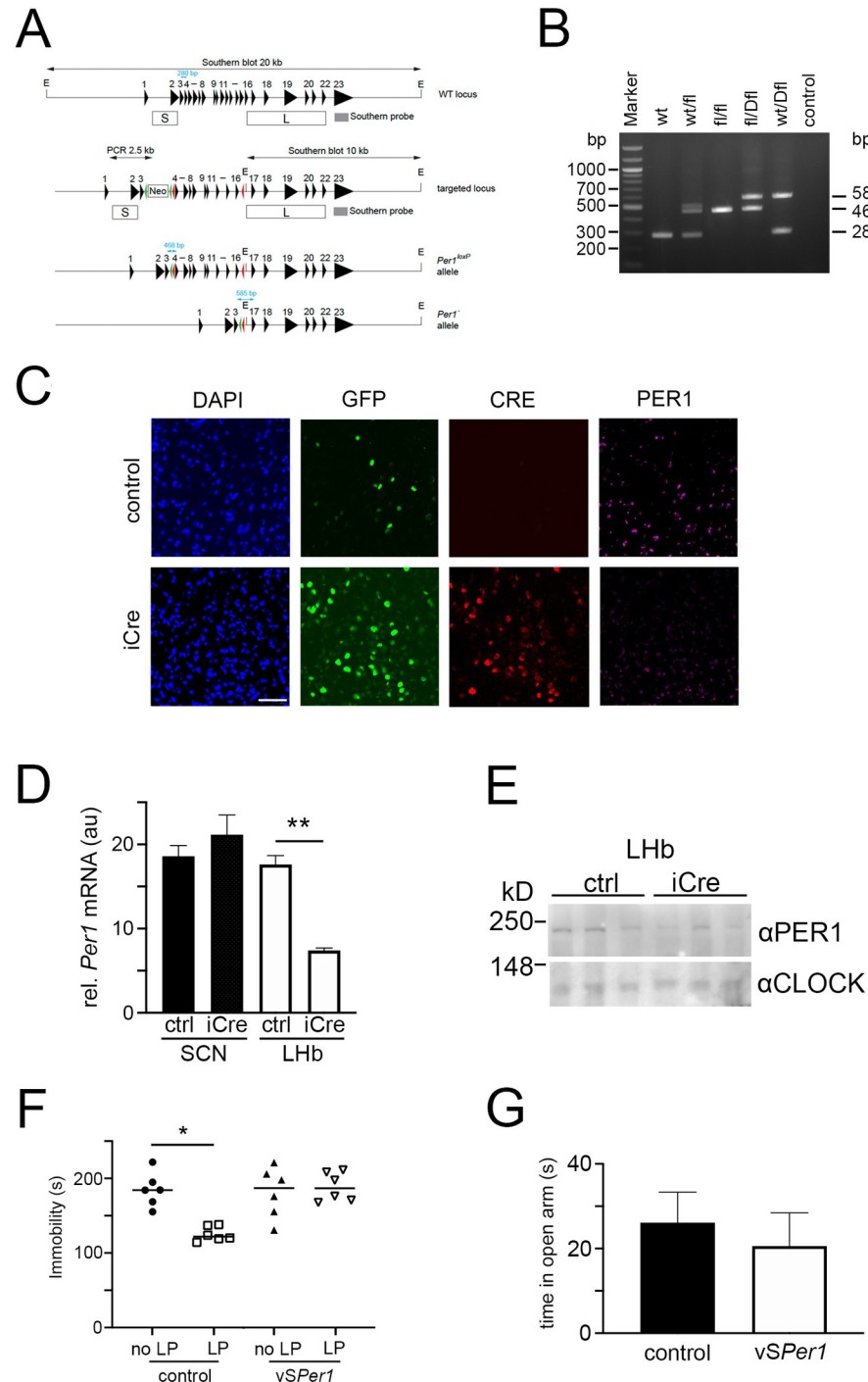

**Fig 3. Deletion of neuronal *Per1* in the area of the lateral habenula abolishes light effects on despair related behavior.** (A) Scheme illustrating the generation of a conditional *Per1* allele. For details see methods section. Numbers above black triangles indicate exons; E, EcoRI; neo, neomycin resistance; green triangles, *FRT* recombination sites; red triangles, *loxP* recombination sites; box S, short arm of homology; box L, long arm of homology; grey box, Southern probe; blue double arrows with numbers, PCR products representing the wild-type (wt), *Per1*$^{loxP}$ (fl), and the *Per1*$^-$ (Dfl) alleles. (B) 1.5% agarose gel visualizing the PCR products representing the wild-type (wt), *Per1*$^{loxP}$ (fl), and the *Per1*$^-$ (Dfl) alleles. (C) Immunohistochemistry visualizing PER1 (pink) in the LHb region of *Per1*$^{fl/fl}$ mice after injection of a control AAV (top panels) or an AAV expressing CRE (red) (bottom panesl). Cell nuclei are visualized by DAPI staining (blue) and GFP (green) shows viral vector infected cells. Scale bar: 50 μm. (D) Quantification of knock-down efficiency in the LHb compared to SCN. *Per1* expression is significantly reduced by AVV Syn-iCre compared to

control (ctrl). One-way ANOVA with Tukey's post-test, n = 4, **p<0.01. (E) Western blot analysis. Protein extracts from the LHb of control AAV or AAV iCre-injected animals were separated on a 7% SDS-PAGE and PER1 and CLOCK were detected with the respective antibodies. Sizes of marker proteins are indicated on the left. (F) Immobility time without and with a light pulse at ZT22 in the FST of male mice at ZT6. Animals were injected with AAV expressing Cre recombinase under the synapsin promoter (vS*Per1*) or control vector. One-way ANOVA with Dunn's multiple comparisons test, n = 6, *p<0.05 (G) Time in open arm of an O-maze at ZT6. Unpaired t-test, n = 11, p>0.05, values in all experiments are means ± SEM.

transcription [7]. Since the flow of information driven by the light signal goes from the LHb via the VTA to the NAc [19] the gene expression in the NAc has an opposite dynamic in presence or absence of PER1 (Fig 4B).

Next, we compared the genes altered in the different brain regions and genotypes (Fig 4C). First and most strikingly, all of the genes influenced by light in the VTA, irrespective of the genotype, were specific for this region only, with no *Per1* mRNA detectable. *Per1* was not detectable, because we were looking for downstream effects of *Per1* gene induction 10 hours after the light pulse when PER1 protein is high (Fig 4A) and its mRNA was already gone. Second, with the exception of two genes, all the light-affected genes in wild type mice were brain region specific. Conversely, in the absence of *Per1* many genes in the NAc and LHb were detected in the SCN as well (21 and 11, respectively). This indicated that many signaling pathways that respond to light and at ZT8 modulate targets common to several brain regions are modified by PER1. Hence, PER1 appeared to contribute to brain region specific shaping of molecular responses to light.

We evaluated tissue-specific differences between the genotypes by comparing for each brain region of interest the expression between wild type and *Per1*[-/-] mice (see S1–S8 Tables). Most of the genes influenced by light in wild type animals appeared to be directly or indirectly regulated by *Per1*, because only a small fraction of those genes was common to the genes found in *Per1*[-/-] mice after light treatment (Fig 4D). Furthermore, we found in wild type the highest number of genes to be modified in the NAc (490) and the lowest number in the VTA (4). These differences in numbers were probably due to the fact that the signal progressing from the LHb to the NAc is a dynamic process and that ZT8 was the optimal time point to detect changes in the NAc, because PER1 protein levels were high at that time point. Interestingly, the number of light-affected genes was higher in all brain regions of *Per1*[-/-] mice, except for the NAc (Fig 4D). This observation highlights the important function of PER1 on pathways which modulate gene expression in the NAc at ZT8.

We were most interested in the 490 genes that were changed by light at ZT8 only in the NAc of wild type mice and appeared to be depending on *Per1* gene expression. Some of these genes are likely involved in the light-mediated effects on behavioral despair that we observed (Fig 1C). We performed a TopGo analysis of this gene set in order to relate these genes to biological processes and molecular functions (Fig 4E). Interestingly, G protein-coupled receptor signaling and sensory perception of smell were the highest-ranking biological processes. This finding was mirrored in the molecular function analysis where G protein-coupled receptor activity and olfactory receptor activity ranked highest (Fig 4E). This result is in line with previous observations that showed depression-like behaviors in rats after olfactory bulb removal [24]. Interestingly, also patients with major depression exhibited olfactory deficits [25].

## Discussion

A growing body of evidence supports the notion that light therapy can be efficient in the treatment of individuals with seasonal and non-seasonal depression [26–32], while light deprivation can increase depressive-like behavior in various species [33–36]. However, many

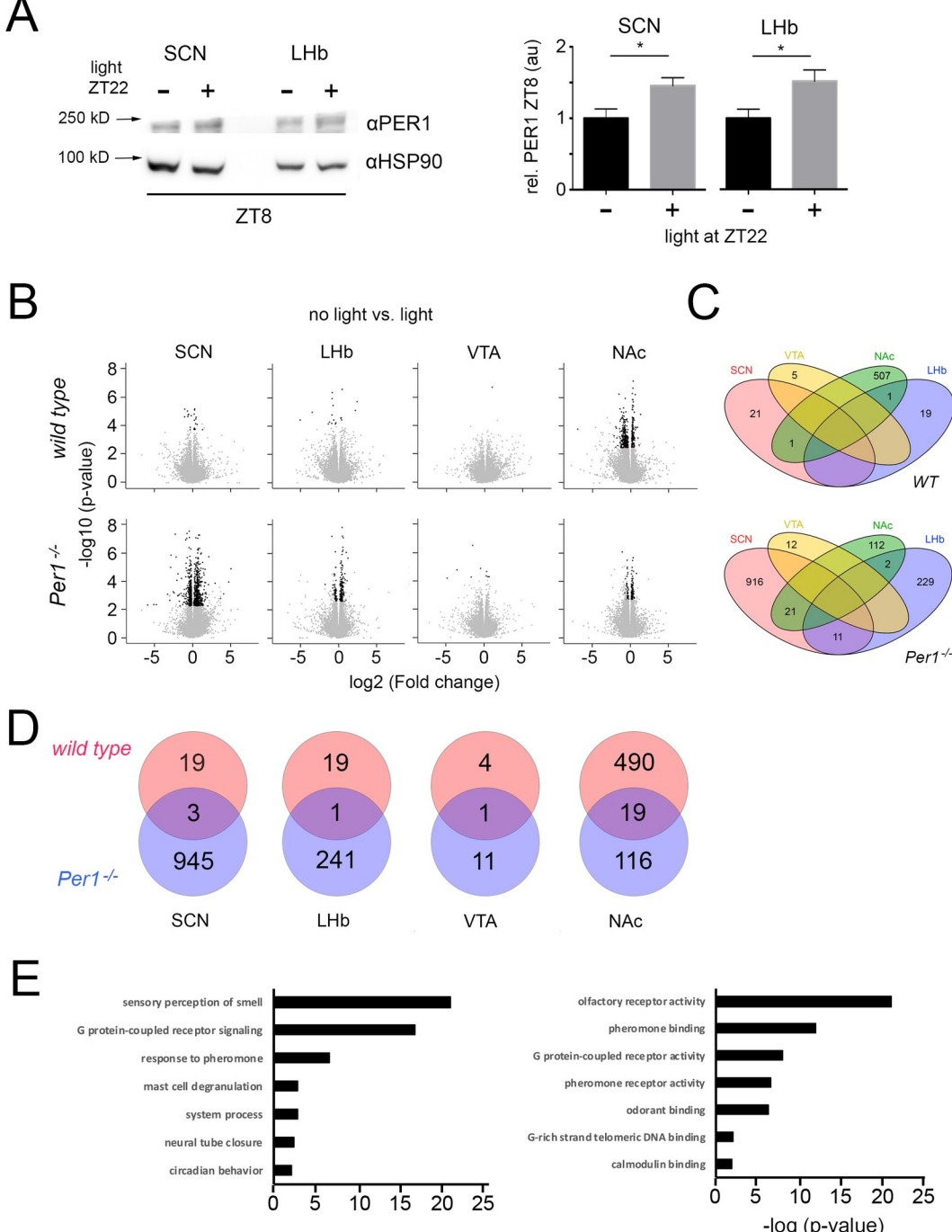

**Fig 4. Light induction of PER1 protein at ZT8 and effects on the transcriptome in the brain.** (A) Expression of PER1 protein at ZT8 in the SCN and the LHb after a light pulse at ZT22 compared to no light pulse. The left panel depicts an example of a Western blot and the right panel shows the quantification of three independent experiments. Unpaired t-test, n = 3, *p<0.05, values are means ± SEM. (B) Volcano plots depicting the changes of gene expression in response to a light pulse at ZT22 in the indicated tissues of wild type and *Per1*[-/-] mice. The RNA sequencing was perfomed 10 hours after the light pulse (ZT8). Black dots indicate significant changes, n = 6. (C) Venn-diagrams illustrating the overlap of gene expression changes in the four nuclei. Top comparison in wild type animals, bottom comparison in *Per1*[-/-] mice. Numbers indicate the amount of genes affected by the light pulse. For the gene names see S1 Table. (D) Venn-diagram comparing wild type and *Per1*[-/-] mice in a single nucleus. Numbers indicate number of genes affected by the light pulse. Genes are listed in S1–S9 Tables; SCN: wt (S2 Table), *Per1*[-/-] (S3 Table); LHb: wt (S4 Table), *Per1*[-/-] (S5 Table); VTA: wt (S6 Table), *Per1*[-/-] (S7 Table); NAc: wt (S8 Table), *Per1*[-/-] (S9 Table). (E) TopGO analysis of biological processes (left) and molecular functions (right) in the NAc of wild type animals.

unresolved questions remain regarding how light therapy can produce its beneficial effects. In this study, we offer evidence that light-mediated induction of the clock gene *Per1* in the LHb is involved in the anti-depressant effects of light therapy.

We studied the effect of light on behavioral despair in mice and uncovered a significant contribution of the clock gene *Per1* in this process. Similar to the observations in humans, where light exerts powerful effects on mood and cognition [12–14], we observed that light at night affected mice in a comparable way (Fig 1A), although mice are nocturnal and not diurnal. We found that light affected behavior in a despair-based paradigm when given in the late part of the dark phase (ZT22). On the other hand, we did not observe any change in this behavior when light was given in the early period of the dark phase (ZT14), as has been reported for rats [17]. This is very similar to light treatment being more efficient in the early morning than evening for patients with seasonal affective disorder (SAD) (reviewed in [11]). Furthermore, we found that light treatment at ZT22 appeared to be more effective in female than male mice (Figs 1A and S1A). However, in humans such a sex-related discrepancy wasn't observed in patients treated for SAD with light [37,38]. Although the incidence of depression seems to be higher in females compared to males [39]. Of note is that the testing of mice in the FST was in their rest phase, whereas in humans it is in the activity phase. This puts a limitation on our interpretations.

Interestingly, we could see the light effect when we assessed animals at ZT6 but not at ZT18. This may indicate that either the light-dark transition at ZT12 may have an impact on the behavioral outcome, or that the circadian clock modulates the amount of immobility in the FST. The latter is more likely, because without any light pulse immobility at ZT18 is lower than at ZT6 in the FST ([40], Fig 1C). Consistently, the lack of *Per1* not only resulted in the absence of a light response but also in an increased immobility time (Fig 1C). Furthermore, it has been shown that the LHb was not activated by a dark-light transition [19].

Mood-related behavior is a complex trait. Therefore, in addition to despair, we tested sucrose preference and behavior in the elevated O-maze, which relate to anhedonia and anxiety, respectively. The results indicated that wild type and *Per1*$^{-/-}$ mice are similar in their ability to experience pleasure (tasting of sucrose) (S1B Fig), but *Per1*$^{-/-}$ mice appeared to be significantly more anxious (Fig 1D and 1E). Hence, *Per1* is likely involved in the despair and anxiety aspects of mood-related behaviors. Altogether, these results support the notion that *Per1* participates in the regulation of mood-related behaviors.

In mammals, light at night can induce gene expression in the SCN [41]. The clock gene *Per1* is among these light inducible genes [9,10]. Since the ipRGCs not only project to the SCN, but also to other brain regions [42], we investigated the habenula [19] for light-inducible *Per1* gene expression. We observed that *Per1* was expressed in the lateral (LHb) and medial habenula (MHb) (Fig 2A), while *Per2* was expressed at a much lower level in these structures (Fig 2B). *Per1* gene expression was induced by light in the LHb when applied at ZT22 (Fig 2C), but not when applied at ZT14 (S2A Fig). These results indicated a time-specific inducibility of *Per1* in the LHb, which mirrored the effect of light at ZT22 observed in the FST as immobility (Fig 1A). Our findings suggest that light induction of *Per1* in the LHb and probably in other brain regions played an important role (Fig 2D). Of note is that *Per2* was most likely not involved, because its gene induction by light at ZT22 was minimal or absent (Fig 2D). Interestingly, light at ZT22 elicits phase-advances, which was abrogated in mice lacking *Per1* but not *Per2* [43]. Hence lack of phase advance and increased immobility in the FST of *Per1* knock-out mice inversely correlated with the observation in humans in which advance of sleep phase had a positive effect on depression [11]. Interestingly, the daily amount of sleep was not different between control and *Per1* knock-out animals (S5 Fig), indicating that the light response in the FST was not simply due to a more awake state of the *Per1* knock-out mice. Taken together,

these results suggested that the amount of *Per1* gene expression in the SCN and the LHb correlate with despair-based behavior as observed in the FST (Fig 1C).

In order to challenge the hypothesis that the amount of *Per1* gene expression in the LHb was relevant for the level of immobility time in the FST, we deleted *Per1* by injecting AAVs expressing iCre into the area of the LHb of *Per1*<sup>fl/fl</sup> mice (Fig 3C–3E). We observed no increase of immobility in the FST in contrast to *Per1* knock-out animals (Figs 1C and 3F), which suggested that *Per1* expression in the LHb is not important for immobility time in the FST. Hence, the oscillator properties of the LHb [44] most likely are not involved in mood related behaviors. However, we observed that the response to a light pulse is abolished when *Per1* is deleted in neurons of the LHb (Fig 3F). This indicated that the light signal leading to a change of immobility in the FST is depending on *Per1* in the LHb. Hence, the LHb clock may contribute to the gating of the light induction of *Per1*.

The importance of the LHb area in mood regulation has been described previously [45]. Light modulated LHb activity via M4-type melanopsin-expressing retinal ganglion cells (RGCs) thereby regulating depressive-like behaviors [19]. Our observations support these findings, although we do not know whether *Per1* is induced in the LHb via the M4-type melanopsin-expressing RGCs. Interestingly, another study described light effects via intrinsically photosensitive RGCs on the SCN and the perihabenula (PHb), regulating hippocampal learning or mood, respectively [18]. Although our experimental set up was different from that study, our findings are not contradictory, because we cannot exclude an involvement of the PHb in our study. The knock-down of *Per1* in the area of the LHb had no effect on anxiety (Fig 3G) compared to the *Per1* knock-out (Fig 1D and 1E). This highlights that *Per1* in other brain regions or body tissues contributed to the phenotype as well. Overall, our results and data by others support the notion, that the area of the LHb and expression of *Per1* are involved in the light mediated effects on mood-related behavior.

The molecular mechanisms through which light elicits its beneficial effects on mood-related behaviors are poorly understood. The observation that light induces the expression of the *Per1* gene in the area of the LHb provides an opportunity to get a first glimpse at potential molecular pathways that are initiated by the activation of *Per1*. In order to identify targets of PER1 that are involved in the behavior we observed in the FST, we performed RNA sequencing experiments. Since clock genes regulate behavior in the FST partly via the mesolimbic dopaminergic system [40,46–48] we used tissue from the LHb, VTA, and NAc with SCN tissue as control. The tissues were isolated at ZT8, because light-induced PER1 protein was highest at that time in the SCN [21] and LHb (Fig 4A) and very close to our behavioral assessment in the FST at ZT6. We observed that light-modulated genes were mostly specific to a particular tissue with virtually no overlap with other tissues assessed. In contrast, lack of *Per1* increased the overlap of light-affected genes between the tissues (Fig 4C), suggesting a role of PER1 in regulating common light-responsive genes. This evidence is consistent with the known role of PER1 as a suppressor of clock and tumor-related genes [7,49]. Interestingly, this can only be observed in the SCN, LHb and VTA, but not the NAc (Fig 4D). This could be due to indirect regulation of genes in the NAc by PER1 (e.g. suppression of suppressors specific to the NAc) and/or neurotransmitter related gene regulation in the NAc by the VTA or other brain regions. Remarkably, the VTA showed the fewest number of genes affected by light. The reason for this may be the time of assessment, because at ZT8, most of the light-modulated genes in the VTA may have already been silenced again. This highlights the highly dynamic nature of light effects on behavior from initial gene expression changes, activation of signaling pathways to neuronal communication and neurotransmitter release. In order to understand this process better, a dynamic assessment of the transcriptome at several time points (e.g. every two hours after the light pulse) would be necessary. The dynamic temporal change of the

transcriptome in the LHb, VTA and NAc after the light pulse may then provide insights into the relationships between the various nuclei to translate the initial light signal into a systemic change that affects mood-related behaviors. Nevertheless, our analysis of the light-responsive transcriptomic change at ZT8 revealed significant contributions of genes involved in the sensory perception of smell/olfactory receptor activity and G protein-coupled receptor signaling/ activity. This parallels previous findings that described depression-like behaviors in rats after olfactory bulb removal [24] and patients with major depression displaying olfactory deficits [25]. However, it remains elusive how the olfactory system and depression are mechanistically related.

Integrated genome-wide association and hypothalamic eQTL studies indicated a link between *Per1* and coping behavior in humans [50]. Furthermore, a single nucleotide polymorphism in the human *Per1* promoter, as well as animal experimentation, revealed a role of *Per1* in psychosocial stress-induced alcohol drinking [51]. These studies are consistent with our finding that *Per1* was involved in the regulation of behavioral despair and anxiety, two aspects of depression. Interestingly, the profiles of *Per1* and *Rev-erbα* were advanced in patients in the manic phase compared with those in the depression phase [52]. This correlates with the light-induced phase-shifting properties of *Per1* when light was applied at ZT22 and eliciting a phase-advance in activity and gene expression rhythms [9,10,43]. Phase shifts of the circadian clock involve the SCN, which are intact in our vS*Per1* mice and still express *Per1* (Fig 3D). Therefore, these animals would phase advance normally. If a phase-advance of the cock would be the only reason for the anti-depressant effects of light at ZT22 then deletion of *Per1* in neurons of the LHb would not be sufficient to abolish the light mediated effects in the FST. Therefore, it is very likely that the effects of light we observe in the FST are due to *Per1* induction in the LHb rather than due to SCN mediated phase advances of the circadian clock.

Taken together, our study provides evidence that the benefit of light on mood-related behaviors involves the clock gene *Per1* and that induction of this gene in neurons in the area of the LHb plays an important role.

**Key Resources Table**

| Reagent type (species) or resource | Designation | Source or reference | Identifiers | Additional information |
|---|---|---|---|---|
| Cell line (M. musculus) | NIH/3T3 | ATCC | Cat. #: ATCC CRL-1658 | Immortalized Mouse fibroblast cells |
| Antibody | anti-PER1 (Rabbit polyclonal) | Created by J. Ripperger [53] | | 1:1000 (WB) 1:100 (IF) |
| Antibody | anti-Clock (Rabbit polyclonal) | Created by J. Ripperger [54] | | 1:1000 (WB) |
| Antibody | anti-HSP90 α/β (mouse, monoclonal) | Santa Cruz Biotechnology | Cat#: sc-13115 RRID:AB_627758 | 1:1000 (WB) |
| Antibody | anti-GFP (Rabbit polyclonal) | Abcam | Cat. #: ab6556 RRID: AB_305564 | 1:500 (IF) |
| Antibody | Anti-Cre recombinase monoclonal antibody (GT10212) | Thermo Fisher Scientific | Cat. #: MA5-27870 RRID: AB_305564 | 1:500 (IF) |
| Antibody | Alexa Fluor 488- AffiniPure Donkey Anti-rabbit IgG (H+L) (Donkey polyclonal) | Jackson Immunoresearch | Cat. #: 711-545-152 RRID: AB_2313584 | 1:500 (IF) |
| Antibody | Alexa Fluor 568- AffiniPure Donkey Anti-rabbit IgG (H+L) (Donkey polyclonal) | Thermo Fisher Scientific | Cat. #: A10037 RRID: AB_2534013 | 1:500 (IF) |
| Antibody | Alexa Fluor 647- AffiniPure Donkey Anti-mouse IgG (H+L) (Donkey polyclonal) | Jackson Immunoresearch | Cat. #: 711-605-151 RRID: AB_2340863 | 1:500 (IF) |

*(Continued)*

| Antibody | anti-mouse IgG–peroxidase, produced in rabbit | Sigma-Aldrich | Cat #: A9044 RRID: AB_258431 | 1:10000(WB) |
|---|---|---|---|---|
| Antibody | anti-rabbit IgG–peroxidase, produced in goat | Sigma-Aldrich | Cat#: A9169 RRID: AB_258434 | 1:10000(WB) |
| Chemical compound | 2% Lidocain HCl | Bichsel | Zul Nr. 53570089 | |
| Chemical compound | Rimadyl (Caprofenum 50mg/ml) | Zoetis | Swissmedic: 54375018 | |
| Chemical compound | RTI-13951-33 hydrochloride | MedChemExpress | Cat. # HY-112612A | |
| Chemical compound | cOmplete, EDTA-free Protease Inhibitor Cocktail | Roche Diagnostics GmbH | Cat. # 06538282001 | |
| Commercial assay, kit | RNeasy Micro Kit | Qiagen | Cat. # 74004 | |
| Commercial assay, kit | SuperScript II Reverse Transcriptase | Invitrogen | Cat. # 18064022 | |
| Commercial assay, kit | KAPA PROBE FAST qPCR Master mix | Kapa biosystems | Cat. # KR0397 | |
| Commercial assay, kit | Lipofectamine 2000 Transfection Reagent | Thermo Scientific | Cat. # 11668019 | |
| Commercial assay, kit | Pierce Rapid Gold BCA Protein Assay Kit | Thermo Scientific | Cat. # A53225 | |
| Commercial assay, kit | Pierce ECL Western Blotting Substrate | Thermo Scientific | Cat. # 32106 | |
| Genetic reagent (M. musculus) | mPer1Brdm1 | Zheng et al., 2001 | | PMID: 11389837 |
| other | DAPI | Thermo fisher | Cat. #: D3571 RRID: AB_2307445 | (1 µg/mL) |
| Recombinant DNA reagent | Adeno-associated viral vectors (AAV) | Viral Vector Facility (VVF) of the Zurich Neuroscience Centre (ZNZ) | v77 (different serotypes) v81-6 v229-6 | |
| Recombinant DNA reagent | Primers and TaqMan probes | Complete list provided in the paper | | |
| Software | Leica application Suite Advanced Fluorescence | Leica | Version 2.7.3.9723 | |
| Software | Prism | GraphPad | Version 7.0d | |
| Software | Quantity one | Biorad | Version 4.6.2 | |
| Software | ImageJ | ImageJ | Version 1.51n | |
| Software | Circ Wave | Roelof A. Hut | Version 1.4 | |

## Materials and methods

### Ethics statement

**Animals and housing.** Animals were kept in 12h light/12h dark (12:12 LD) cycle with food and water *ad libitum*. Timing of experiments is expressed as zeitgeber time (ZT, ZT0 lights on, ZT12 lights off). Up to 4 animals were housed in plastic cages (268mm long x 215mm×141 mm high; Techniplast Makrolon type 2 1264C-001) covered with a stainless-steel wire lid (Techniplast 1264C-116) and a filter top (1264C-400SC; 1264C-420R). Unless otherwise stated, both male and female mice were used in experiments. Housing and experimental procedures were performed in accordance with the guidelines of the Schweizer Tierschutzgesetz and the declaration of Helsinki. The state veterinarian of the Canton of Fribourg approved the protocol (2015-33-FR). *Per1^Brdm1^* total knock out animals [55] were used.

**Generation of conditional *Per1* mice (*Per1^loxP^*).** A cassette containing *neomycin acetyltransferase (neo)* flanked by two FRT sites [56] was inserted into a bacterial artificial chromosome encompassing the *Per1* locus (clone bMQ-339m04). Two loxP sites flanking exon 4–16 of *Per1* were introduced to delete exons 4–16 and the reading frame could be resumed in exon

22, thereby deleting almost the entire gene. The neo cassette for selection in embryonic stem cells was between exon 3 in the short arm of homology (1.4 kB) and the loxP site upstream of exon 4. The loxP site downstream of exon 16 was followed by the long arm of homology (4 kB) containing exons 17–22. 1x10$^7$ ES cells (HM-1) were electroporated with the linearized targeting vector using standard methods. Gene targeting was performed by PolyGene (Rümlang, Switzerland). Homologous recombination was verified using a primer just outside the short arm of homology of the targeting vector (5'-CTGCCTTTCCTGTCACTATC-3') and a primer in the neo-cassette (5'-GTTGTGCCCAGTCATAGCCGAATAG-3') resulting in an amplicon of 2.5 kb. Recombination on the long arm of homology was verified by Southern blotting using the following primers to generate a probe of 490 bp in size: (5'-TGTGGCAGCAGCGTT CAAG-3') and (5'-GGCGTGGACAATCCTCCAAATG-3'). The probe recognized a 10 kb band for the targeted allele and a 20 kb band for the wild-type allele. C57Bl/6 female mice (Charles River, WIGA Sulzfeld) were super-ovulated using standard procedures and mated with C57Bl/6 breeder males (Charles River). Blastocysts were injected with ES cell clones and transferred into pseudo-pregnant B6CBAF1 females (Polygene, Rümlang, Switzerland, originally obtained as inbred strains from Harlan Laboratories). Male chimeras were bread with C57Bl/6 females. The offspring was then screened for the presence of the neo gene, since the females did not contain the Flp recombinase and hence targeted animals still contained the neo cassette. The primers used to detect the neo cassette were: Neo.MP1: 5'-GCTGTGCTCCA CGTTGTCAC-3' (neo-cassette and Chromosome 3)

Neo.MP5: 5'-GGAAAGCTGGGCTTGCATCTC-3' (Chromosome 3)

Neo.MP6: 5'-GGAGCGGCGATACCGTAAAG-3' (neo-cassette)

The amplicons for Neo were 512 bp and for wild-type 380 bp. The targeted animals with the neo cassette were then bread with *Flp* deleter mice to remove the *neo* cassette [57]. Offspring were tested for deletion of the neo cassette using the following primers:

F001.1: 5'-GAGCAGGACAACCCATCTAC-3'

F001.22: 5'-ACCCTGAACCTGCTTGAC-3'

giving rise to a wild-type amplicon of 280 bp and a targeted amplicon of 468 bp. To confirm the presence or absence of the Flp-recombinase we use the following primers:

SD24: 5'-CTAATGTTGTGGGAAATTGGAGC-3'

SD25: 5'-CTCGAGGATAACTTGTTTATTGC-3'

the Flp-allele was 568 bp. Screening for the distal loxP site was done with the following primers:

F001.23: 5'-GGAGAGCTGCAACATTCC-3'

F001.24: 5'-GGAGCTGAAGCTACACTGAC-3'

F001.25 (loxP): 5'-ACTAGTTCTAGAGCGGCCGAGC-3'

The primer pairs gave rise to the following amplicons: F001.23/F001.24 wild-type allele 476 bp, targeted allele 723 bp. F001.24/F001.25 only detected the targeted allele with an amplicon of 358 bp.

For detection of the deleted allele after Cre-mediated recombination we used the primers F001.1 and F001.24 (see above) giving rise to an amplicon of 585 bp.

**Stereotaxic injections.** The following adeno-associated viruses (AAV) were provided by the Viral Vector Facility (VVF) of the Neuroscience Center Zürich (ZNZ): for serotype testing—v77: ssAAV-X/2-hCMV-chI-EGFP-WPRE-SV40p(A) where X indicates the serotype: either 1, 2, 5, 6, 8, 9, or DJ were used. For injections into the LHb of floxed/floxed *Per1* mice the following AVV vectors were used: as control, v81-6: ssAAV-6/2-hSyn1-EGFP-WPRE-hGHp(A) with a titer of 9.3 x 10$^{12}$ viral genome equivalents/ml; for Cre delivery, v229-6: ssAAV-6/2-hSyn1-EGFP_iCre-WPRE-hGHp(A) with a titer of 6.8 x 10$^{12}$ viral genome equivalents/ml. AVV vectors and plasmids required for their synthesis are available on the VVF website (https://www.vvf.uzh.ch/en.html).

2-months old mice were selected for the stereotaxic injections. Under isoflurane/oxygen anesthesia (2.0% isoflurane v/v, 1 l $O_2$/min, induction: 5.0% v/v, 1.0l $O_2$/min) the animal's head was fixed in the stereotaxic instrument (Kopf instruments) and fitted with an anesthesia inhalation mask (Stoelting). The depth of anesthesia was periodically monitored with the toe-pinch reflex. Bregma was used as reference point for positioning the skull and coordinates (position 0). The brain was accessed with craniotomy. The injection (200 nl at 40 nl/min) at LHb-specific coordinates (AP = -1.75mm, ML = +/-0.9mm, DV = 2.7mm, inward angle +/-10°) was performed with a pulled glass pipette (Drummond, 10µl glass micropipette, Cat number: 5-000-1001-X10) attached to a hydraulic manipulator (Narishige: MO-10 one-axis oil hydraulic micromanipulator). The micropipette was risen by 0.1mm and kept in place for 5 min to prevent upward spread of the virus. For analgesia 2% lidocain was applied locally and 5 mg/kg of caprofen was injected subcutaneously on the day of the surgery and during the two following days. Animals that underwent the surgery were left in their home cage for 14 days to recover.

**Light pulse experiment.** Mice aged 3–6 months were exposed to polychromatic light at 500 lux in their home cage for 30 min at ZT14 or ZT22 (until ZT22.5). The control group was subjected to the same handling (cage movement, presence of the experimenter) in the dark but receiving no light pulse.

**Behavioral testing.** To reduce stress and burden for the animals gentle handling methods were applied wherever possible, including cup and tunnel handling [58,59]. For behavioral testing, mice were caged individually at least 4 days before the experiment.

*Forced swim test (FST)* was performed according to established protocols [20] at an experiment-specific zeitgeber time. A transparent cylinder 35 cm high with a diameter of 25 cm was filled with water up to 20 cm high. The temperature of the water was kept at 25 +/- 1˚C. During a 6-minute session the animal was placed in the cylinder and allowed to swim freely while the experimenter left the room to minimize disturbance. The session was recorded horizontally with a video camera and the video was scored manually. During one session, two animals were tested at the same time in neighboring cylinders separated by a thin white wall to obstruct the view. The test result was expressed as immobility time in seconds recorded during the last 4 minutes of the test. Small movements of paws and tail meant to stabilize the animal's position while floating were not considered as mobility. Scoring was performed blinded with the assessor not knowing the genotype or treatment of the animals.

Anxiety was measured using *elevated O-maze* test at ZT6. Mice were placed on an elevated ring platform (40 cm Ø), divided into four parts, equal in size, two exposed, and two flanked by 15 cm high walls. Each animal underwent a single 10-minute session which was recorded by video camera from above. Time in the exposed part, entries into the exposed portion (full body), head dips and stretch-attend postures were counted manually. Scoring was performed blinded with the assessor not knowing the genotype or treatment of the animals.

**Sucrose preference test.** Sucrose (1, 2.5, and 5% w/v) solution intake was measured in a two-bottle free choice test (sucrose against water) [60]. After weighing, mice were placed into individual cages at least three days before the start of the experiment. One day before the beginning of the experiment mice were habituated to the presence of two bottles in the cage both containing water. After 12 h the consumed water for each bottle was measured and the position of the bottles was inverted. At the start of the test, one water bottle and one bottle containing a sucrose solution (1, 2.5 or 5% w/v) were introduced to each cage. Each sucrose concentration was present for three days). The amount of consumed water or sucrose solution in 24 h was measured with exchanging the position of the bottles every 12 h. Sucrose preference was calculated by dividing the amount of sucrose consumed by the total amount of consumed solutions (water and sucrose). Mice were weighed every day.

**Brain region isolation.**   A freshly isolated brain was cut on a brain matrix. The regions of interest were cut out of the corresponding brain slice and immediately frozen in liquid nitrogen. Tissue was stored in -80˚C until further analysis.

**Immunostaining.**   Brain tissue was washed with 0.9% NaCl, 10KU/l Heparin and subsequently fixed with 4% PFA by cardiovascular perfusion. The tissue was cryoprotected with 30% sucrose/1xPBS (137 mM NaCl, 7.97 mM $Na_2HPO_4 \times 12$ $H_2O$, 2.68 mM KCl, 1.47 mM $KH_2PO_4$). Coronal sectioning was performed on a cryostat (Microm, Thermo Scientific, HM550) at 30 μm section thickness. Pre-selected slices were washed for 5 min once in 1xTBS (0.1 M Tris/0.15 M NaCl) and twice in 2xSSC (0.3 M NaCl/0.03 M tri-Na-citrate pH 7). Antigen retrieval comprised of 50 min incubation in 65˚C in 2xSCC/50% formamide. Subsequently, the slices were washed 2 x 5 min in 2xSSC, 3 x in 1xTBS and blocked for 1 h at 25˚C in 10%FBS/1xTBS/0.1%Triton. Primary antibody (anti-GFP 1:500, Abcam, ab6556; or a combination of anti-PER1 (1:100) and anti-Cre (1:500)) was diluted in 1%FBS/1xTBS/0.1% Triton and incubated for 18h at 4˚C. Next the sections were washed 3 x with 1xTBS. Secondary antibody (Alexa Fluor 488 AffiniPure Donkey Anti-Rabbit IgG (H+L), 1:500 in 1%FBS/1xTBS/0.1% Triton, or Alexa Fluor 568 AffiniPure Donkey Anti-Rabbit IgG (H+L) and Alexa Fluor 647 AffiniPure Donkey Anti-Mouse IgG (H+L) 1:500 in 1%FBS/1xTBS/0.1% Triton) was added for a 3 h incubation at 25˚C, followed by 3 washes in 1xTBS, 10 min DAPI staining (1:5000 in PBS; Roche) and 2 washes with 1xTBS. The slices were mounted on slides (SlowFade antifade reagent, Invitrogen) and visualized with a confocal microscope.

Confocal pictures were taken using a Leica TCS SP5 microscope with a 10x or 63x objective. Resolution of the image was set to 1024x1024 pixels, scanning speed to 200 Hz with 20% laser power. Images were captured using Leica LAS (2.7.3) software and processed using ImageJ (1.51n).

**Quantitative polymerase chain reaction (qPCR) analysis.**   RNA from tissue fragments was isolated with RNeasy Micro Kit (Qiagen) according to manufacturer's instructions. Reversed transcription was performed using SuperScript II Reverse Transcriptase (Invitrogen) according to the manufacturers protocol (250–1000 ng RNA per reaction). After reverse transcription, samples were diluted 10 times. 5μl of the diluted sample was added to 7.5μl of KAPA mix (KAPA PROBE FAST qPCR Master) and 2.5 μl of primers/probe mix (listed in S10 Table) (150 nM of primers, 33.3 nM TaqMan probe). Samples were analyzed using the Rotor-Gene qPCR machine (Corbett research, RG-6000). For the primers used see S10 Table.

**Transcriptome analysis.**   Female mice (4.5 months of age) were sacrificed at ZT8 (10 h after a 30 min light pulse at ZT22) and brain regions were isolated as described above. For RNA isolation, two animals were pooled for each sample (total of 12 animals analyzed in 6 separate samples) for each phenotype and treatment group (wild-type, wild-type plus light pulse, *Per1*-/-, *Per1*-/- plus light pulse) to match a total of 24 samples per brain region. RNA isolation was performed using the RNeasy Micro Kit (Qiagen) following the manufacturers instructions. 500 ng of each sample was converted into a sequencing-ready library using the CATS RNA-seq kit with rRNA depletion (Diagenode). The RNA integrity, effectiveness of the rRNA depletion and the final library sizes were verified by running aliquots on a TapeStation 2200 system with the appropriate ScreenTape devices (Agilent). Then, 12 samples for each brain region were pooled to equal amounts. The samples were sequenced on a total of 8 lanes using a HiSeq3000 instrument (Illumina) with a single end flow cell for 75 cycles. The obtained sequencing reads were demultiplexed with bcl2fastq version 2.20 with default parameters. Raw sequence data were pre-processed with a local Galaxy server (version 17.09), where FASTQ Groomer (version 1.1.1) and FASTQ Trimmer (version 1.1.1) were run [61]. The quality of the sequence data was checked by FastQC (version 0.11.8) [62]. The pseudoalignment with Kallisto (version 0.45.1) [63] was performed with the sequence data and an index built from the

mouse transcriptome (Mus_musculus.GRCm38) with K-mer size = 21. The resulting Abundance.h5 files were imported to R (version 3.6.0) for differential gene expression analysis with the library DESeq2 [64] and gene enrichment analysis with topGO [65]. The detailed software setup and programming scripts can be found on Github (https://github.com/kayihui/effect_of_light_on_brain_regions). The data have been deposited in the SRA database under the accession number PRJNA628975. For the SCN and NAc we used the data from six animals each, for the LHb one light-induced sample was lost due to too low reads, and for the VTA four animals each (two samples were found to be too severely contaminated with another brain region).

**Cell culture, protein isolation for shRNA screening.** NIH 3T3 mouse fibroblast cells (ATCCRCRL-1658) were maintained in Dulbecco's modification of Eagle's medium (DMEM, with 4.5 g/l glucose, L-glutamine & sodium pyruvate, Corning) with fetal bovine serum (FBS, 10% v/v, Biowest) and Penicillin/Streptomycin (100 U/mL, Roche Diagnostics GmbH). Cells were cultured on 10 cm sterile plates in 37°C, in humidified a chamber with 5% $CO_2$. Lipofectamine 2000 (Invitrogen) was used to transfect cells (at 70% confluence, one day after seeding) according to manufacturer's protocol. 10 μg of plasmid DNA (4 variants of shRNA carrying plasmids, OriGene TL501619) DNA was used to transfect one plate. The medium was replaced 6 hours after transfection with a standard one. 3 days after transfection cells were washed twice with cold PBS (137 mM NaCl, 7.97 mM $Na_2HPO_4 \times 12 \ H_2O$, 2.68 mM KCl, 1.47 mM $KH_2PO_4$), detached in PBS and moved to a 1.5 ml tube. After centrifugation the cell pellet was lysed in 1 ml of cold RIPA buffer (50 mM Tris-HCl pH7.4, 1% NP-40, 0.5% Na-deoxycholate, 0.1% SDS, 150 mM NaCl, 2 mM EDTA, 50 mM NaF) with freshly added protease (cOmplete ULTRA tablets, EDTA-free, Roche Diagnostics GmbH) and phosphatase inhibitors (PMSF 1mM, $Na_3VO_4$ 0.2mM). The protein isolate was separated from cell debris by centrifugation (15 min, 14000g, 4°C) and stored at -20°C.

**Protein isolation from brain tissue.** Isolated brain regions from 3 animals were lysed in 500 μl cold lysis buffer (50 mM Tris-HCl, 150 mM NaCl, 0.25% SDS, 0.25% Sodium Deoxycholate, 1 mM EDTA) with freshly added protease (cOmplete ULTRA tablets, EDTA-free, Roche Diagnostics GmbH) and phosphatase inhibitors (PMSF 1 mM, $Na_3VO_4$ 0.2 mM). Protein isolates were separated from cell debris by centrifugation (15 min, 14000 g, 4°C) and stored at -20°C.

**Western blot.** Protein sample concentration was determined by BCA (Pierce Rapid Gold BCA Protein Assay Kit). Samples were mixed with Laemmli buffer, incubated for 5 min at 95°C and 80 μg of protein was separated on a 10% SDS-PAGE gel (alongside PageRuller Plus Prestained protein Ladder, Thermo Scientific). Samples were transferred onto a nitrocellulose membrane (Amersham Protan 0.45 μm, GE healthcare). Blocking was performed at 25°C for 1 h (5% milk, 1xTBS, 0.1% Tween), followed by 16 h primary antibody incubation at 4°C (rabbit anti-PER1 antibody, 1:1000; rabbit anti-CLOCK, 1:1000; in 5% milk, 1xTBS, 0.1% Tween). Next, the membrane was washed 3 times with 1xTBS/0,1% Tween, incubated for 3 h at 25°C with the secondary antibody (anti-rabbit IgG peroxidase 1:10000, Sigma-Aldrich; anti-mouse IgG peroxidase 1:10000, Sigma-Aldrich; in 5% milk, 1xTBS,0.1% Tween) and washed 3 times with 1xTBS/0.1% Tween. After a 5 min incubation with the developing reagent (Pierce ECL Western Blotting Substrate, 32106, Thermo Scientific) the images were captured using an Azure 300 imaging system (Azure Biosystems).

**In situ hybridization.** Animals were sacrificed at the experiment specific zeitgeber time. Tissue preparation, sectioning, $\alpha^{35}$S-UTP labeled riboprobe production and hybridization was followed according to [66].

*mPer1* and *mPer2* probes were produced from pBluescript SK(-) and pCRII-TOPO respectively. The specificity of the antisense probe was verified using sense probe for hybridization.

The signal obtained from the X-ray film (Amersham Hyperfilm MP, GE Healthcare) was quantified using densitometric analysis (GS-800, BioRad with Quantity One software, Biorad). In the case of weak signal, quantification was performed using silver-stained dark-field microscopy pictures (Zeiss Axioplan) coupled with Hoechst staining for nuclei. For both methods, the relative probe signal was calculated by subtracting the background signal taken from the neighboring region.

### Sleep assessment and analysis

Sleep-wake behavior was recorded and analyzed using the non-invasive PiezoSleep system (Signal Solutions LLC, Lexington, KY) based on a highly sensitive piezo based motion detector system [67]. Mice were housed individually for 13 days within the PiezoSleep cages with bedding, nesting material and food and water *ad libitum*. Prior to the sleep recording, mice were allowed to habituate to their new environment for two days. Baseline sleep was then recorded for six consecutive days without any intervention. One day later the 30 min light pulse was applied at ZT22 and sleep recording was continued for the following four days. Final numbers of mice were n = 6 for each genotype (*WT* and *Per1*$^{-/-}$). One mouse from each genotype had to be excluded as the piezo decision algorithm failed to distinguish properly between the wake and sleep state due too much bedding/nesting material interfering with breathing rate measurement.

Using the PiezoSleep analysis software (v2.0), the data collected from the cage system was binned over one hour to obtain the average amount of sleep per hour during baseline sleep. For sleep bout length distribution analysis to determine sleep consolidation, the data recorded during a period of two hours between ZT6 –ZT8 before and after the light pulse (average of day 2, 3 and 4 post LP) was binned over 4 seconds. Each bout was attributed to one of eight logarithmically increasing bin lengths of >4-8s, >8-16s, >16-32s, >32-64s, >64-128s, >128-256s, >256-512s and >512s. Additionally to the frequency distribution, the amount of sleep the mice spent sleeping in each of the bout duration categories was computed [68].

**Statistical analysis.**   Depending on experimental design, an appropriate method of statistical analysis was used (two-way ANOVA, one-way ANOVA, student t-test). Those statistics were calculated using GraphPad Prism software version 7.0d. A p-value lower than 0.05 was considered statistically significant. Circ Wave 1.4 [69] was used for cosinor analysis of expression patterns.

## Supporting information

**S1 Fig.**  (A) Immobility time in the forced swim test (FST) of wild type male mice assessed over several days at ZT6 after no light pulse (LP) (black bars), after a LP at ZT14 (red bars), and after a LP at ZT22 (blue bars). Two-way repeated measures ANOVA (n = 13–15), ZT14 LP p = 0.44, ZT22 LP p = 0.14, values are means ± SEM. (B) Sucrose preference was tested allowing mice to choose between water or sucrose (1–5% (weight/vol)) over 3 days for each sucrose concentration. Two-way RM ANOVA revealed no differences between wild type (WT, n = 21, black lines) and Per1$^{-/-}$ mice (n = 18, grey lines), p = 0.67, values are means ± SEM. (TIF)

**S2 Fig.**  (A) Light induction of Per1 at ZT14 in the LHb and the SCN. Dark-field images of coronal sections containing the habenula (rostral to caudal, left panels) and the SCN as positive control (right panels). The yellow signal represents the hybridization signal detecting Per1 mRNA and blue represents Hoechst-dye stained cell nuclei. The MHb can be distinguished from the LHb by the densely packed blue-colored nuclei. Brain section of mice sacrificed 60 min. after the light pulse (bottom row) and the corresponding controls are shown (top row).

Scale bar: 200μm. (B) Brain region specific markers for verification of isolated brain regions used for further analysis. Quantitative PCR comparing various genes in the SCN, LHb, VTA and NAc. The most specific gene for the SCN is irs4, for the LHb it is gpr151, for the VTA it is tacr3 and for the NAc it is tac1. One-way ANOVA with Tukey's multiple comparisons test was used, n = 12–16, ****p<0.0001, values are means ± SEM.
(TIF)

**S3 Fig.** (A) Brain sections of the habenular region are shown. Optimization of infection efficiency was performed by testing different variants of adeno-associated virus (AAV) expressing GFP (green color). Blue indicates cell nuclei (DAPI staining). For the lateral habenula (hatched white lines) AAV6 appeared to have the most localized and strongest infection potential. Scale bar: 200 μm. (B) Tissue controls for PCR deletion verification of *Per1* in Fig 3C. The *Irs4* mRNA is only detected in SCN tissue and not the LHb (left panel). The *Gpr151* mRNA is only detected in LHb tissue and not the SCN (right panel).
(TIF)

**S4 Fig. Brain region-specific markers for verification of isolated brain regions used for further analysis.** Quantitative PCR comparing various genes in the SCN, LHb, VTA and NAc. The most specific gene for the SCN is irs4, for the LHb it is gpr151, for the VTA it is tacr3 and for the NAc it is tac1.
(TIF)

**S5 Fig. Sleep in wild-type (WT) and *Per1*$^{-/-}$ mice.** (A) Amount of sleep in % over 24 hours under a 12-hour light and 12-hour dark cycle. n = 6 per genotype. Two-way ANOVA shows that the two curves are not significantly different. (B) Number of sleep bouts per duration bin, before (left) and after (right) a light pulse (LP) at ZT22. n = 6 per genotype, no differences are observed. (C) Amount of sleep in % in the different sleep bout bins. Before the light pulse (LP) (left) a significant difference can be seen in the sleep bouts >128s and >512s between the two genotypes. This difference vanishes after the light pulse (right panel). n = 6, t-test, *p < 0.05.
(TIF)

**S1 Table. Genes depicted in Fig 4C.**
(XLSX)

**S2 Table. Genes for wild type SCN shown in Fig 4D.** In red the genes common with the genes of *Per1*$^{-/-}$ SCN.
(XLSX)

**S3 Table. Genes for *Per1*$^{-/-}$ SCN shown in Fig 4D.** In red the genes common with the genes of wild type SCN.
(XLSX)

**S4 Table. Genes for wild type LHb shown in Fig 4D.** In red the genes common with the genes of *Per1*$^{-/-}$ LHb.
(XLSX)

**S5 Table. Genes for *Per1*$^{-/-}$ LHb shown in Fig 4D.** In red the genes common with the genes of wild type LHb.
(XLSX)

**S6 Table. Genes for wild type VTA shown in Fig 4D.** In red the genes common with the genes of *Per1*$^{-/-}$ VTA.
(XLSX)

**S7 Table. Genes for *Per1*<sup>-/-</sup> VTA shown in Fig 4D.** In red the genes common with the genes of wild type VTA.
(XLSX)

**S8 Table. Genes for wild type NAc shown in Fig 4D.** In red the genes common with the genes of *Per1*<sup>-/-</sup> NAc.
(XLSX)

**S9 Table. Genes for *Per1*<sup>-/-</sup> NAc shown in Fig 4D.** In red the genes common with the genes of wild type NAc.
(XLSX)

**S10 Table. Primers used for RT qPCR analysis.**
(DOCX)

# Acknowledgments

We thank Antoinette Hayoz, Stéphanie Aebischer, Jean-Charles Paterna (Viral Vector Facility, University of Zürich), Doron Schmerling (PolyGene, Rümlang) and the Bioimage platform (University of Fribourg) for technical support.

# Author Contributions

**Conceptualization:** Jürgen A. Ripperger, Urs Albrecht.

**Data curation:** Jürgen A. Ripperger, Ka Yi Hui.

**Formal analysis:** Iwona Olejniczak, Jürgen A. Ripperger, Federica Sandrelli, Laureen Mansencal-Strittmatter, Katrin Wendrich, Ka Yi Hui.

**Funding acquisition:** Urs Albrecht.

**Investigation:** Iwona Olejniczak, Jürgen A. Ripperger, Federica Sandrelli, Anna Schnell, Laureen Mansencal-Strittmatter, Katrin Wendrich, Ka Yi Hui, Andrea Brenna, Naila Ben Fredj, Urs Albrecht.

**Methodology:** Iwona Olejniczak, Jürgen A. Ripperger, Federica Sandrelli, Anna Schnell, Laureen Mansencal-Strittmatter, Ka Yi Hui, Andrea Brenna, Naila Ben Fredj.

**Project administration:** Urs Albrecht.

**Resources:** Urs Albrecht.

**Software:** Ka Yi Hui.

**Supervision:** Urs Albrecht.

**Validation:** Iwona Olejniczak, Jürgen A. Ripperger, Federica Sandrelli, Anna Schnell, Laureen Mansencal-Strittmatter, Andrea Brenna, Naila Ben Fredj.

**Visualization:** Ka Yi Hui, Urs Albrecht.

**Writing – original draft:** Urs Albrecht.

**Writing – review & editing:** Iwona Olejniczak, Jürgen A. Ripperger, Federica Sandrelli, Anna Schnell, Laureen Mansencal-Strittmatter, Andrea Brenna, Urs Albrecht.

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
