## [Decision Letter · Decision Letter 0]

10 Jun 2020

Dear Dr Albrecht,

Thank you very much for submitting your Research Article entitled 'Title: Light affects behavioral despair involving the clock gene Period 1' to PLOS Genetics. Your manuscript was fully evaluated at the editorial level and by independent peer reviewers. The reviewers appreciated the attention to an important problem, but raised some substantial concerns about the current manuscript. Based on the reviews, we will not be able to accept this version of the manuscript, but we would be willing to review again a much-revised version. We cannot, of course, promise publication at that time.

If you decide to revise the manuscript for further consideration at PLOS Genetics, please aim to resubmit within the next 60 days, unless it will take extra time to address the concerns of the reviewers, in which case we would appreciate an expected resubmission date by email to plosgenetics@plos.org.

[LINK]

We are sorry that we cannot be more positive about your manuscript at this stage. Please do not hesitate to contact us if you have any concerns or questions.

Yours sincerely,

Achim Kramer

Associate Editor

PLOS Genetics

Gregory Barsh

Editor-in-Chief

PLOS Genetics

Reviewer's Responses to Questions

**Comments to the Authors:**

Reviewer #1: In this manuscript, the authors found that light pulses at ZT22 change behaviors that are usually releated to depression- and anxiety. Furthermore, they found that a light pulse given at this time activates the expression of the circadian clock gene Period1 (Per1) in the lateral habenula (LHb). A total knockout and a knockdown of PER1 in the LHb leads to depression-like behavior. Therefore, the authors conclude that Per1 in the LHb might play an important role in positive effects of light therapies in depression. Additionally, the authors show that a light pulse during the second half of the dark phase causes profound changes of gene expression in the suprachiasmatic nucleus, the LHb and the nucleus accumbens (NAc) in WT mice, which were different in Per1-/- mice.

This is an interesting study of high relevance, which aims to link clinical treatment strategy to molecular mechanism. The manuscript is well-written and the figures are clearly presented. All methods the authors used are standard. However, the interpretation of the results is not very conclusive and there are some major and minor concerns that need to be considered before publication.

Major:

1) Since the authors use untreated WT mice, it might not be appropriate to address their levels of immobility in the FST test as signs of despair or depression, unless there is reason to believe that despair and depression are default traits of mice. Instead of measuring immobility, which can occur due to many reasons, it might be more meaningful to test susceptibility to develop depression-like behavior in response to stress.

2) In this context, it is also not clear whether the light pulse at ZT22 really changes depression-like behavior. It is also possible that due to the light pulse, the mice are simply more awake (and therefore more mobile) when tested a few hours later.

3) The interpretation that Per1 in the LHb mediates positive behavioral effects of light pulses on mood cannot be based on the presented data. To come to this conclusion, the authors would need to test behavior after a light pulse in ShPer1 mice.

4) The presented RNA-based analyses are rather descriptive and the interpretation are rather speculative (e.g. the role of Per1 in the suppression of inhibitors in the NAc and the connection between genes involved in the olfactory sense and the olfactory bulb in depression).

Minor:

a) Fig 3C shows immobility of shPer1 mice while Fig 1C shows immobility of total Per1-/- mice. It would be much more intuitive to see how much of the effects were contributed by loss of Per1 in LHb if results of WT, Per1-/- and shPer1 mice were shown and analyzed together.

b) It might make sense if the authors could also show PER1 protein level of the LHb of ShPer1 mice especially when they found the knock-down of Per1 did not supress all Per1 mRNA.

c) In Fig 4A, the differences of the western blot bands are not clearly visible.

d) The conclusion that Per1 is involved in the perception of reward is not in line with the findings of this study, as no change in the sucrose preference was detected.

e) In line 219, the word “we” is missing.

Reviewer #2: The authors explore the contribution of Per1 to the anti-depressant/anxiolytic effect of light and probe specific brains for their involvement. The authors found that a light pulse at ZT22 but not at ZT14 reduced time spend immobile in the FST when performed at ZT6 but not at ZT18 and that Per1 null mice were more immobile at any timepoint independent of light. Elevated O-maze mirrored this result with Per null spending less time spend in the open.

The authors found evidence for Per1 rhythms in the LHb and further found that a ZT22 light pulse upregulated Per1 expression in the lateral/medial Hb. Induction was also found in the VTA and Nacc but to a smaller extend.

Next the authors knocked down Per1 in the LHb and found increased immobility in the FST. Notably the authors also found elevated PER1 in the LHb 10hr (at ZT8) post light pulse. Finally, the authors conducted RNA Seq on brains extracted 10hrs post +/- light pulse at ZT22 and found strong diff gen expression in the Nacc in WT while this was attenuated in Per1 nulls while SCN and LHb showed profound increase in diff express genes in Per nulls. Interestingly, the VTA did not show much change in either.

This is an interesting manuscript adding to the emerging importance of the habenula and its circadian properties in mood regulation. The experiments are carefully controlled pointing to a role of the LHb local clock or local Per1 gene expression in mood regulation. Specifically intriguing is the robust RNA Seq result indicating that Per1 either in the SCN and/or the habenula suppresses pathways that in turn suppress NAc ‘activation’ by light which might be key to the antidepressant effect of light. Removal of Per1 therefore consequently leads to Nac ‘inactivation’. This finding is very compelling as the Nacc is a key player in mood regulation and the (perhaps exclusive) target of psychostimulant action.

There are a few aspects of the data presentation and discussion that need work.

The fact that Per1 is a core clock component and at the same time light inducible should be related to the findings, how can the findings be explained by an effect on the clock (phase advance…)? Is there room for an explanation based on light induction only, on pathways independent of the clock? Could it be a mix of both?

The result in Fig4A needs more interpretation: could the increase in PER1 levels at ZT 8 suggest an advance of the PER peak meaning that it is a consequence of phase shift rather than a ‘plastic’ long-term change?

In this regard the long lasting effect of a single light pulse on FST (5days!) is hard to fathom…what could possibly explain such sustained effect?

RE RNA Seq, the volcano plots clearly show changes for both up and down regulated genes, this needs to be more carefully addressed (e.g the term ‘induction’ is incorrect because many genes are becoming suppressed), perhaps use ‘diff regulated’.

The lack of response in the VTA is interesting, could it be that the SCN/LHb acts on the Nacc without the VTA as mediator? The Nacc is innervated by many regions not only the VTA. The current explanation given for the lack of response to light of the VTA transcriptome is not clear/too hand-wavy to me.

The posture result in Fig.3 is very significant, perhaps repeat in global Per-/- and add to figure 1? Not critical but would strengthen the knock down approach!

**Have all data underlying the figures and results presented in the manuscript been provided?**

Reviewer #1: Yes

Reviewer #2: Yes

PLOS authors have the option to publish the peer review history of their article (what does this mean?). If published, this will include your full peer review and any attached files.

Reviewer #1: No

Reviewer #2: No

---

## [Decision Letter · Decision Letter 1]

9 Feb 2021

Dear Dr Albrecht,

Thank you very much for submitting your Research Article entitled 'Title: Light affects behavioral despair involving the clock gene Period 1' to PLOS Genetics.

The manuscript was fully evaluated at the editorial level and by independent peer reviewers. The reviewers appreciated the attention to an important problem, but raised some substantial concerns about the current manuscript. Based on the reviews, we will not be able to accept this version of the manuscript, but we would be willing to review a much-revised version. We cannot, of course, promise publication at that time.

If you decide to revise the manuscript for further consideration at PLOS Genetics, please aim to resubmit within the next 60 days, unless it will take extra time to address the concerns of the reviewers, in which case we would appreciate an expected resubmission date by email to plosgenetics@plos.org.

[LINK]

We are sorry that we cannot be more positive about your manuscript at this stage. Please do not hesitate to contact us if you have any concerns or questions.

Yours sincerely,

Achim Kramer

Associate Editor

PLOS Genetics

Gregory Barsh

Editor-in-Chief

PLOS Genetics

Reviewer's Responses to Questions

**Comments to the Authors:**

Reviewer #1: I am sorry to hear that the authors had problems with their control vector. I give them all the more credit for the extensive work they did to answer the questions from us reviewers. The paper is much improved and I don't see any other issues that need to be worked on.

Reviewer #2: In this revision, the authors replaced a key element of the paper, the shRNAi knock down of Per1 in the LHb, with a Cre loxP approach, and explain this with an issue with the control vector under conditions of light induction. It is not clear why they removed all of the shRNAi results, as for instance the increased FST immobility of the knock down mice was a result independent of light, where the empty/scrambled vector had no effect but the shRNAi construct did. If there is no concern with the data, the authors should retain this shRNAi result (increased immobility).

Regarding the new model, demonstration of selective disruption (the authors should use such term and not ‘knock down’ in the context of Cre loxP) is not convincing. The IHC in Fig. 3 appears rather unspecific. Shouldn’t PER1 be predominantly nuclear? There seem to be no hint of any nuclear staining. Has the antibody been validated? How does it look in a WT versus Per1-/- SCN? The authors need to show higher magnification images preferably co-labeled with a neuronal marker to demonstrate that the IHC is specific (side-by-side with Per1-/- sections). Alternatively the authors may want to employ FISH (RNAscope) to demonstrate efficacy of the disruption. A count of all PER1+ cells (co-labeled with a neuronal maker) in the ROI (=LHb) of selective disrupted and WT mice would be important to quantify the efficacy. Also, a lower mag image would be helpful to demonstrate the spatial specificity of Cre action.

The IHCs of the AAV-GFPs in Fig. S3A indicate that even AAV6 seems only to affect half of the LHb…so a careful assessment of disruption efficacy is very much needed.

The authors still fail to acknowledge that light does not simply induce transcripts but affects the transcriptomes fairly evenly in both directions for all tissues and genotypes, if one uses the volcano plots as guides.

In this context, the identification of differentially expressed transcripts needs explanation. The cut off p-values below which a transcriptional change is considered significant seems to vary between tissues and genotypes according the black labeled dots in the volcano plots, this needs clarification.

With regard to the NAc the authors use the term induction of transcript thereby giving the impression as if light leads to ‘activation’ of the Nacc 10 hrs later. However, the volcano plot does not allow for such interpretation as induced and suppressed transcript seem to roughly equal in number (the suppressed transcripts seem to be even higher, gauging form the volcano plots).

Because of this, the notion that Per1 acts as a suppressor is not supported by the transcriptomics data, rather Per1’s role in the LHb is to prevent ANY action of light on the LHb transcriptome which would otherwise lead to negative action onto despair circuits. Given that Per1Ko prevents even downregulation of transcripts, Per1 might have a higher order function here e.g. preventing light from accessing any aspect of transcriptional regulation.

In general I am not convinced that one should engage in interpretation of the RNAseq data with regard to depression-related behaviors and circuits/regions without a detailed look at individual transcripts (and perhaps more time points), especially given the fact that light seem to suppress as many transcripts as it is activating. At this point it seem unclear what the transcriptional changes actually mean: activation or silencing of the circuits? Or none of the above? The pathway analysis provided is really very coarse and clearly incomplete as it does not address at all the numerous downregulated transcripts in the Nacc and elsewhere.

Given that the LHb/Habenula is considered one of the most rhythmic extra-SCN brain oscillators (see e.g. Guilding Piggins et al. 2007), the authors should discuss a possible contribution of Per1 through the local LHb clock.

Finally, the correlation/reference to SAD is perhaps overreaching, given that mice are nocturnal and thus much of the effects of light on mood might be actually inverse. Evidence for this is provided by light masking, which is completely inverse and can be viewed as inverse arousal regulation. And arousal is relevant, as one of the hall marks of SAD is sleep problems, specifically difficulty to fall asleep during periods of winter gloom. Also, light therapy improves mood during the daytime, however, the ZT22 light pulse (2hours before ‘wake up’) has no effect on the ‘daytime’ mood of mice (ZT18), ZT6 is really a mid-sleep time point. In this context, Huang et al. (Neuron 2018) did apply ‘light therapy’ during the daytime (ZT1-3) and over a course of 10/14days. Regardless, using a nocturnal model for SAD is clearly a challenge and that needs to be acknowledged more in the discussion.

Minor:

-The abstract does not take into account the new model, selective disruption of Per1 in the LHb.

-“At this time point, light not only causes eventual adaptation of the mammalian circadian clock to a new time zone but is also most effective in the treatment of some forms of depression, such as seasonal affective disorder [11].”

Is this true, has light at ZT22 (=2hrs before light on i.e. before wake up) been used for SAD treatment?

**Have all data underlying the figures and results presented in the manuscript been provided?**

Reviewer #1: Yes

Reviewer #2: Yes

PLOS authors have the option to publish the peer review history of their article (what does this mean?). If published, this will include your full peer review and any attached files.

Reviewer #1: No

Reviewer #2: No

---

## [Decision Letter · Decision Letter 2]

24 May 2021

Dear Dr Albrecht,

Thank you very much for submitting your Research Article entitled 'Light affects behavioral despair involving the clock gene *Period 1*

' to PLOS Genetics.

The manuscript was fully evaluated at the editorial level and by independent peer reviewers. The (remaining) reviewer appreciated the attention to an important topic but identified one last (minor) concern that we ask you address in a revised manuscript before we can accept it. 

We therefore ask you to modify the manuscript according to the review recommendations. Your revisions should address the specific points made by the reviewer.

[LINK]

Yours sincerely,

Achim Kramer

Associate Editor

PLOS Genetics

Gregory Barsh

Editor-in-Chief

PLOS Genetics

Reviewer's Responses to Questions

**Comments to the Authors:**

Reviewer #2: The authors addressed all comments to my satisfaction, except for one:

My point:

Evidence for this is provided by light masking, which is completely inverse and can be viewed as inverse arousal regulation. And arousal is relevant, as one of the hall marks of SAD is sleep problems, specifically difficulty to fall asleep during periods of winter gloom. Also, light therapy improves mood during the daytime, however, the ZT22 light pulse (2hours before ‘wake up’) has no effect on the ‘daytime’ mood of mice (ZT18), ZT6 is really a midsleep time point.

And the authors response:

The light pulse in the early morning in SAD treatment is most efficient in the dark phase (ZT22) in humans. The later the pulse occurs the less efficient the treament in humans (Wirz-Justice et al., ref 11 in the paper). The beneficial effect on mood is in humans as well as in mice is in the light phase. It is correct that for humans this is the activity phase and in mice the rest phase.

My comment:

The authors really need to explain why an improvement at ZT6 is relevant and speaks to the human situation as it would equate to waking up a human in the middle of the night to conduct the behavioral test, which is not something that is typically done in humans.

Thus the correct time for testing would seem to be the wake period (ZT12-ZT24), not ZT6.

The authors should discuss this limitation as it speaks also more broadly to the meaningfulness of behavior testing in nocturnal rodents during day time to learn something relevant for the human situation.

Also with regard to the same clock phasing in nocturnal and diurnal animals: this is true for the SCN but it seems that the output is quite immediately converted, suggesting that the arousal rhythmicity that spreads across the brain is really very much antiphasic in such species.

**Have all data underlying the figures and results presented in the manuscript been provided?**

Reviewer #2: Yes

PLOS authors have the option to publish the peer review history of their article (what does this mean?). If published, this will include your full peer review and any attached files.

Reviewer #2: No

---

## [Editor Report · Decision Letter 3]

27 May 2021

Dear Urs,

We are pleased to inform you that your manuscript entitled "Light affects behavioral despair involving the clock gene *Period 1*" has been editorially accepted for publication in PLOS Genetics. Congratulations!

Yours sincerely,

Achim

Achim Kramer

Associate Editor

PLOS Genetics

Gregory Barsh

Editor-in-Chief

PLOS Genetics

Comments from the reviewers (if applicable):

**Data Deposition**

http://datadryad.org/submit?journalID=pgenetics&manu=PGENETICS-D-20-00731R3

**Press Queries**

---

## [Editor Report · Acceptance letter]

16 Jun 2021

PGENETICS-D-20-00731R3 

Light affects behavioral despair involving the clock gene *Period 1*

Dear Dr Albrecht, 

We are pleased to inform you that your manuscript entitled "Light affects behavioral despair involving the clock gene *Period 1*" has been formally accepted for publication in PLOS Genetics! Your manuscript is now with our production department and you will be notified of the publication date in due course.

With kind regards,

Zsofi Zombor

PLOS Genetics

On behalf of:
